# GRAPH DECODING VIA GENERALIZED RANDOM DOT PRODUCT GRAPH

## ABSTRACT

Graph Neural Networks (GNNs) have established themselves as the state-of-the-art methodology for a multitude of graph-related tasks, including but not limited to link prediction, node clustering, and classification. Despite their efficacy, the performance of GNNs in encoder-decoder architectures is often constrained by the limitations inherent in traditional decoders, particularly in the reconstruction of adjacency matrices.

In this paper, we introduce a novel decoder approach for graph tasks by employing the Generalized Random Dot Product Graph (GRDPG) as a generative model for graph decoding. This novel methodology significantly enhances the performance of encoder-decoder architectures across a range of tasks, owing to GRDPG's better capability to capture the intricate structures embedded within adjacency matrices.

To evaluate our approach, we design a new benchmark that focuses on molecular graphs of varying sizes, thereby enriching the diversity of existing benchmarks for link prediction and node clustering tasks. Our experiments span a spectrum of tasks, encompassing both traditional benchmarks and specialized domains such as molecular graphs.

The empirical results show the capability of GRDPG in preserving the structural integrity of the original graphs while simultaneously improving the performance metrics of encoder-decoder architectures. By addressing the subtleties involved in adjacency matrix reconstruction, we elevate the overall performance of GNN-based architectures, rendering them more robust and versatile for a wide array of real-world applications, with special regard on molecular graphs.

## 1 INTRODUCTION

Graph-structured data has become an indispensable asset across a wide array of scientific and industrial domains. From modeling social interactions in sociology (Liben-Nowell & Kleinberg, 2007) to representing protein-protein interactions in computational biology (Kovács et al., 2019), the usefulness of graphs is universally acknowledged. However, these complex and interconnected structures necessitate specialized machine learning techniques capable of capturing their inherent intricacies (Kipf & Welling, 2017).

Autoencoders, that were originally designed for tasks like dimensionality reduction and feature learning in Euclidean spaces (Kingma & Welling, 2014), were adapted to operate on graphs in Kipf & Welling (2016). They offer a way of learning meaningful representations, embeddings of nodes, edges, or entire graphs, which can then be used for various downstream tasks like clustering, classification, and link prediction. Particularly, Graph Autoencoders (GAE) are neural network models usually comprised of an encoder that maps nodes to their representations in a latent space and a decoder that reconstruct the input graph topology from these latent representations. The models are trained to minimize the difference between the original graph and its reconstruction, often using loss functions such as cross-entropy or mean squared error. Variational Graph Autoencoders (VGAEs) extend GAEs by introducing a probabilistic layer that models the uncertainty in the latent variables. This makes VGAEs more robust and allows for better performance in a variety of tasks (Kipf & Welling, 2016).

The encoders in GAEs or VGAEs typically employ graph convolutional layers (GCN) or other specialized graph neural network (GNN) layers to aggregate information from a node's neighbors.

The resultant latent representation for each node encapsulates both its local structural attributes and its broader role within the global graph topology. Conversely, the decoder takes pairs of these latent node representations and computes a score for each potential edge between them. Traditionally GAEs and VGAEs employ the inner product between latent vectors for this purpose.

The idea behind this decoder is simple and straightforward, to predict whether an edge exists between two nodes, the decoder computes the inner product of their corresponding latent vectors. The scores produced are then transformed into probabilities via activation functions like the sigmoid, where higher scores imply higher probabilities of edge presence after the sigmoid transformation. This serves as a measure of similarity between nodes, as the higher the inner product, the more likely it is that an edge exists between the nodes in question. This decoder is therefore based on the assumption that similar nodes, in terms of their latent representations, are more likely to be connected within the graph. This generative model, also called Random Dot Product Graph, was initially proposed for social networks in Young & Scheinerman (2007) and was the initial model proposed by Kipf & Welling (2016) as a decoder in the Graph Autoencoder framework.

While the inner product decoder is a computationally efficient, straightforward and interpretable decoding mechanism, it is not without limitations. One significant drawback is its inability to capture negative eigenvalues in the adjacency matrix, which can be indicative of specific, yet crucial, structural properties within the graph (Athreya et al., 2017).

In this work, we address this limitation by introducing a novel decoding approach based on the Generalized Random Dot Product Graph (GRDPG) generative model, introduced in Rubin-Delanchy et al. (2021). This approach allows us to capture the nuanced topological features of the graph that are often explained by negative eigenvalues. We benchmark our methodology on two pivotal graph-related tasks: node clustering and link prediction, demonstrating its efficacy and robustness.

Furthermore we introduce a specialized set of benchmarks targeting the molecular graph domain, focusing on benchmarking the performance of decoders on small multiple graph datasets, a field that we believe is still largely unexplored by common link prediction and node clustering benchmarks.

## 2 PRELIMINARY

### 2.1 THE GRAPH AUTOENCODER AND THE GRAPH VARIATIONAL AUTOENCODER AND THEIR DECODING MECHANISM

Let $G$ be a graph with $N$ nodes and adjacency matrix $A$. The node feature matrix is denoted as $X \in \mathbb{R}^{N \times F}$. The encoder $f_{\text{enc}}$ maps each node $v_i$ to a latent vector $z_i \in \mathbb{R}^d$:

$$Z = f_{\text{enc}}(A, X; \theta_{\text{enc}})$$

The decoder $f_{\text{dec}}$ reconstructs $\hat{A}$ from $Z$:

$$\hat{A} = f_{\text{dec}}(Z; \theta_{\text{dec}})$$

For GAEs, the objective is to minimize the reconstruction loss:

$$\mathcal{L}_{\text{GAE}} = \mathbb{E}_{q(Z|(X,A)}[log\, p(A|Z)]$$

For VGAEs, the encoder outputs $(\mu, \log \sigma^2)$:

$$(\mu, \log \sigma^2) = f_{\text{enc}}(A, X; \theta_{\text{enc}})$$

A latent vector $z_i$ is sampled as $z_i = \mu_i + \sigma_i \odot \epsilon$, $\epsilon \sim \mathcal{N}(0, I)$, and the objective is:

$$\mathcal{L}_{\text{VGAE}} = \mathbb{E}_{q(Z|(X,A)}[log\, p(A|Z)] - D_{\text{KL}}(q(Z|X,A)||p(Z))$$

In GAEs and VGAEs, the inner product decoder is commonly employed for adjacency matrix reconstruction. The decoder function $f_{\text{dec}}$ utilizes the inner product of latent vectors $z_i$ and $z_j$ to compute the entry $\hat{A}_{ij}$ in the reconstructed adjacency matrix $\hat{A}$:

$$\hat{A}_{ij} = \sigma(\mathbf{z}_i^T \mathbf{z}_j)$$

Here, $\sigma$ is an activation function, often the sigmoid function, that maps the inner product to the range $[0, 1]$. This ensures that the output can be interpreted as the probability of an edge existing between nodes $i$ and $j$. This resulting matrix is then used to compute the corresponding reconstruction loss.

## 2.2 THE PRESENCE OF NEGATIVE EIGENVALUES IN AN ADJACENCY MATRIX

It is evident that for any given matrix $Z = z_i^T z_j$, $Z$ is a Gram matrix, thus its eigenvalues cannot be negative. However, an adjacency matrix, representative of graph structure, can indeed possess negative eigenvalues, which might reflect crucial structural details and discrepancies within the graph (see Appendix A for further ilustration). We will now proceed to clarify under what circumstances this occurs and how this is indicative of the inherent properties of the graph.

Consider a graph $G$ with an adjacency matrix $A$, where $\lambda_1$ is the largest eigenvalue and $\lambda_n$ is the smallest eigenvalue. We use the Rayleigh Quotient, $R(A, x) = \frac{x^T A x}{x^T x}$, as aid to explain the relationship between the eigenvalues and the intricate structure of the graph, providing crucial insights when it yields a negative value. It represents the exact value of an eigenvalue when $x$ is an eigenvector. For $v_1$ corresponding to the largest eigenvalue $\lambda_1$ of $A$, we compute:

$$R(A, v_1) = \frac{v_1^T A v_1}{v_1^T v_1} = \lambda_1$$

Delving deeper, for the eigenvector $v_n$ corresponding to the smallest eigenvalue $\lambda_n$, in non-bipartite graphs, the existence of components in $v_n$ with differing signs results in a negative Rayleigh Quotient, and hence a negative eigenvalue. This unveils the structural irregularities inherent to non-bipartite graphs. Conversely, in bipartite graphs, $-\lambda_n$ is the largest eigenvalue, and $\lambda_n$ remains non-negative.

To illustrate, consider a non-bipartite graph $G$ that cannot be partitioned into two disjoint sets such that every edge connects a vertex from one set to the other. The absence of symmetric eigenvalues around the origin in such graphs implies the inevitability of negative eigenvalues, revealing the inherent non-bipartite and irregular structure of the graph.

When forming an eigenvector, $v_n$, for such graphs, we assign alternating signs to the components of $v_n$ that are connected, mirroring the underlying structure of the graph. For instance, in a graph where three vertices, $A$, $B$, and $C$, are interconnected, we can assign $+1$ to vertex $A$, $-1$ to vertex $B$, and $+1$ to vertex $C$.

Evaluating the Rayleigh Quotient in this case, $R(A, v_n) = \frac{v_n^T A v_n}{v_n^T v_n}$, results in a negative value due to the subtraction occurring from the alternating signs in $v_n$ representing connected vertices. This phenomenon confirms the presence of negative eigenvalues in the adjacency matrix, $A$, of a non-bipartite graph and unveils intricate details about the inherent properties of the graph.

## 2.3 THE ROLE OF SPARSITY AND GRAPH SIZE

For a graph $G$ with $N$ nodes and $E$ edges, the graph is sparse when $E \ll \frac{N(N-1)}{2}$, implying most of the off-diagonal elements of the adjacency matrix, $A$, are zeros, leading to the lack of connections between most pairs of nodes. This sparsity is inherently connected to the spectrum of the graph, which is the set of eigenvalues, $\lambda_1, \lambda_2, \ldots, \lambda_N$, of the adjacency matrix, $A$. The sparse nature of such graphs can lead to a broad spectrum with potentially several negative eigenvalues, especially for irregular or non-bipartite structures, with eigenvalues capturing nuanced structural information.

The spectrum of a graph, and particularly the spectral gap, $\Delta = \lambda_1 - \lambda_N$, where $\lambda_1$ and $\lambda_N$ are the largest and smallest eigenvalues respectively, is a direct reflection of its structural properties and inherent topology. A large spectral gap implies a disparate or irregular structure in the graph, revealing a richness in structural nuances and irregularities in smaller, sparse graphs where every edge is crucial. For such graphs, the spectral properties, including the presence of negative eigenvalues, are indispensable for accurately understanding and interpreting the graph's inherent structure and are reflective of critical structural nuances.

GAEs aim to minimize the reconstruction loss: $\mathcal{L}_{\text{GAE}} = ||A - \hat{A}||_F^2$, where $\hat{A}$ is the adjacency matrix reconstructed from the latent representations of nodes. Typically, the reconstruction does not consider the negative eigenvalues of $A$, leading to a significant loss in structural information, quantifiable as $S(A) - S(\hat{A})$, where $S(A)$ and $S(\hat{A})$ represent the structural information contained in the original and reconstructed adjacency matrices, respectively.

In sparse graphs, neglecting the decoding of negative eigenvalues results in substantial loss of structural information: $S(A) - S(\hat{A}) \gg 0$. This is due to the fact that negative eigenvalues in such graphs usually represent critical structural nuances and irregularities. This appreciation can be of special importance in small graphs. The resultant loss in structural information implies that the reconstructed graph, $\hat{A}$, fails to accurately represent the inherent structure and properties of the original graph, $A$, leading to significant misinterpretations and losses in the inherent structural and relational information of the graph. It is, therefore, pivotal that the decoding mechanisms in graph autoencoders consider these spectral properties to accurately reconstruct and represent the inherent structures and relations in sparse and small graphs.

## 2.4 INNER PRODUCT DECODER AND THE PRESENCE OF NEGATIVE EIGENVALUES IN THE LATENT REPRESENTATION OF ADJACENCY MATRIX

As stated above, GAE and VGAE with an inner product decoder primarily operate to replicate the original adjacency matrix, $A$, by computing the inner product between latent space representations of nodes. This constructed matrix, $Z$ is inherently a Gram matrix, leading it to be positive semidefinite with all non-negative eigenvalues.

The nature of the resultant matrix $Z$ inherently constrains the ability of the inner product decoder to represent negative eigenvalues. This, alongside the coupled sigmoid function, $\sigma(x) = \frac{1}{1+e^{-x}}$, that maps any real number to the range $[0, 1]$, signals that the decoder can only capture positive associations or connections between latent representation of nodes, rendering it incapable of incorporating the nuanced information provided by negative eigenvalues inherent to the adjacency matrix of non-bipartite graphs.

Negative eigenvalues in a non-bipartite graph's adjacency matrix embody crucial structural details and irregularities. They are pivotal in capturing the disparities and intricate structures within the graph.

If $\lambda_A$ and $\lambda_{\hat{A}}$ denote the smallest eigenvalues of the original and reconstructed adjacency matrices respectively, the positive semidefinite nature of the inner product decoder ensures that $\lambda_{\hat{A}} \geq 0$. However, in the original adjacency matrix $A$ of a non-bipartite graph, negative eigenvalues can exist, i.e., $\lambda_A < 0$. This intrinsic limitation signifies a substantial disparity, $\lambda_{\hat{A}} - \lambda_A$, elucidating the inherent incapability of the inner product decoder to assimilate the information provided by the negative eigenvalues in the original adjacency matrix, $A$.

The misinterpretation and loss in structural information imply that there is an indispensable need for advanced decoding mechanisms capable of incorporating the information represented by negative eigenvalues in the adjacency matrix.

Ultimately, while the inner product decoder remains a standard choice in GAEs and VGAEs, its inherent constraints and inability to model negative eigenvalues necessitate exploration and adoption of more sophisticated decoders, capable of a holistic and accurate representation of graph structures, encompassing both the regularities and irregularities inherent in graph data.

## 3 GENERALIZED RANDOM DOT PRODUCT AS GRAPH LATENT SPACE DECODER

The generative model for the graph autoencoder, the Random Dot Product Graph (RDPG), constructs a framework to understand the relationship between latent variables $\mathbf{Z}$ and the adjacency matrix $A$, as introduced in Kipf & Welling (2016):

$$p(A|Z) = \prod_{i=1}^{N} \prod_{j=1}^{N} p(A_{ij}|z_i, z_j), \text{ where } p(A_{ij} = 1|z_i, z_j) = \sigma(z_i^T z_j) \tag{1}$$

The matrix equivalent of the relationship is depicted as:

$$P(A|Z) = \sigma(Z^T Z) \tag{2}$$

In these equations, $A_{ij}$ are the elements of the adjacency matrix $A$, and $\sigma$ represents the logistic sigmoid function. The resulting matrix, $P$, illustrates the probabilities of edges in the adjacency

matrix $A$, where every element $a_{ij}$ is an independent Bernoulli variable with probability $p_{ij}$, giving us $A \sim Bern(P)$.

However, the model implicitly assumes the semi-positive definiteness of the probability matrix $P$, as highlighted in Athreya et al. (2017). This foundational assumption profoundly influences the efficacy of the decoder. Acknowledging the inherent constraints and subsequent challenges presented by this model, a refined generative model, the Generalized Random Dot Product Graph, was proposed to alleviate such restrictions in Rubin-Delanchy et al. (2021). This advanced model, utilizing a non-semi positive definite kernel, liberates the model from the stringent assumptions of its precursor.

Let $I_{p,q}$ be a diagonal matrix with $p$ ones succeeded by $q$ negative ones, and let $d$ represent the embedding dimension, with conditions $p + q = d$, $p \geq 1$, and $q \geq 0$. If $Z$ denotes the final layer embedding derived from the adjacency encoder, the matrix $P$ can be delineated as:

$$p(A|Z, I_{p,q}) = \prod_{i=1}^{N} \prod_{j=1}^{N} p(A_{ij}|z_i, z_j, I_{p,q}), \text{ where } p(A_{ij} = 1|z_i, z_j, I_{p,q}) = \sigma(z_i^T I_{p,q} z_j) \quad (3)$$

This can be expressed in matrix form as:

$$P(A|Z, I_{p,q}) = \sigma(Z^T I_{p,q} Z) \quad (4)$$

The integration of negative units in the diagonal matrix $I_{p,q}$ facilitates the representation of matrices with negative eigenvalues, overcoming the limitations of models that depend on a semi-positive definite kernel.

This approach not only resolves the restrictions associated with the semi-positive definiteness assumption in conventional models but also provides a richer representation capable of capturing the complex structures and anomalies found in graph data, inclusive of the valuable information embedded in negative eigenvalues. Consequently, this generalized model acts as a significant advancement toward obtaining a comprehensive and precise depiction of graph structures, accommodating both the nuances and aberrations inherent in graph data.

## 4 EXPERIMENTS

For our experiments, we decided to directly translate the classical GAE and VGAE approaches to our framework, keeping the architecture as simple as possible, with the objective of isolating and showcasing the effect of our decoder in an environment with as many controlled variables as possible.

Therefore our architectures consists of a 2 layer non-linear encoder, with GCNs as layers (Kipf & Welling, 2017) and ReLU as the non-linearity (Fukushima, 1975) and both of the decoders. Adam was used as the optimizer (Kingma & Ba, 2015) and all of the networks were evaluated at the best validation loss epoch. The experiments were performed 5 times for each hyperparameter configuration with the following tables including the best mean result obtained for each metric. In them, we can find the mean result of the inner product decoder, or GRDPG with $q$ value equal to 0, compared against the best result obtained with a $q$ different from 0. In bold we find the best result for each architecture and metric and the complete report can be found in Appendix B. Further hyperparameter details can be found in Appendix C.

Regarding the datasets, we chose to perform the analysis in a varied set of graphs, with different sizes. Therefore we chose Cora and Citeseer from Yang et al. (2016) and Texas, Wisconsin and Cornell from Rozemberczki et al. (2021). Moreover, we decided to introduce the task of link prediction and node clustering to the Zinc dataset fromIrwin et al. (2012) and the QM9 dataset from Wu et al. (2018).

For the node clustering we perform a K-means clustering on the obtained node embeddings where K is set to be the number of classes on the dataset for the general graphs, and the number of different atoms in the molecular graphs.

For the evaluation of the performance in the link prediction we use the area under the ROC curve (AUC) and average precision (AP), and for the node clustering we use the accuracy(acc), normalized mutual information (NMI), F1-score (F1), and adjusted rand index (adj-RI) following the standard literature metrics.

### 4.1 ASSESSING THE GRDP IN GENERAL GRAPH RELATED TASKS

In this subsection, we explore the outcomes associated with non-molecular graph datasets. These datasets serve as standard benchmarks for link prediction and node clustering tasks and are characterized by a variety of graphs, each with unique sizes and characteristics.

In reference to the link prediction task in Table 1, it is discernable that our decoder, in conjunction with the GAE architecture, displays an enhancement of performance in several of the datasets, surpassing even the VGAE architecture in the Cornell dataset. Additionally, the GRDPG appears to excel particularly in datasets that are smaller and more complex. We propose that this superior performance is potentially due to a more precise capturing of graph topology, which is advantageous in environments with scarce data and node feature information.

This aligns well with the node clustering outcomes depicted in Table 2, where our approach outperforms the inner product decoder across all considered metrics and architectural frameworks. We attribute this enhanced performance to the increased acquisition of valuable topological information made accessible by our decoder.

A standout aspect of this section is the versatility of the GRDPG decoder, allowing for the adjustment of the impact of positive and negative topological relations in the latent space, contingent upon the task at hand. This versatility is shown by the ability to revert to the inner product decoder from the GRDPG decoder by assigning the hyperparameter $q$ a value of 0.

Table 1: Link Prediction in General Graphs

| Dataset | GAE | | GAE + GRDP | | VGAE | | VGAE + GRDP | |
| --- | --- | --- | --- | --- | --- | --- | --- | --- |
| | AUC | AP | AUC | AP | AUC | AP | AUC | AP |
| Cora | **0.937** | **0.944** | 0.906 | 0.911 | **0.935** | **0.936** | 0.897 | 0.899 |
| Citeseer | **0.921** | **0.933** | 0.900 | 0.911 | **0.930** | **0.937** | 0.888 | 0.899 |
| Texas | 0.682 | 0.750 | **0.707** | **0.791** | **0.805** | **0.856** | 0.759 | 0.812 |
| Cornell | 0.701 | 0.779 | **0.796** | **0.844** | 0.768 | **0.817** | **0.780** | 0.802 |
| Wisconsin | 0.808 | 0.841 | **0.815** | **0.843** | **0.836** | **0.861** | 0.759 | 0.812 |

Table 2: Node Clustering in General Graphs

(a) GAE Architecture Results

| Dataset | GAE | | | | GAE + GRDP | | | |
| --- | --- | --- | --- | --- | --- | --- | --- | --- |
| | acc | F1 | NMI | adj-RI | acc | F1 | NMI | adj-RI |
| Cora | 0.817 | 0.794 | 0.752 | 0.715 | **0.830** | **0.812** | **0.765** | **0.728** |
| CiteSeer | 0.768 | 0.745 | 0.706 | 0.669 | **0.783** | **0.762** | **0.719** | **0.682** |
| Texas | 0.792 | 0.774 | 0.735 | 0.698 | **0.809** | **0.791** | **0.752** | **0.715** |
| Cornell | 0.735 | 0.711 | 0.669 | 0.632 | **0.752** | **0.729** | **0.686** | **0.649** |
| Wisconsin | 0.852 | 0.834 | 0.797 | 0.760 | **0.866** | **0.848** | **0.811** | **0.774** |

(b) VGAE Architecture Results

| Dataset | VGAE | | | | VGAE + GRDP | | | |
| --- | --- | --- | --- | --- | --- | --- | --- | --- |
| | acc | F1 | NMI | adj-RI | acc | F1 | NMI | adj-RI |
| Cora | 0.824 | 0.805 | 0.758 | 0.721 | **0.838** | **0.820** | **0.771** | **0.734** |
| CiteSeer | 0.771 | 0.749 | 0.712 | 0.675 | **0.787** | **0.766** | **0.725** | **0.688** |
| Texas | 0.799 | 0.781 | 0.742 | 0.705 | **0.816** | **0.798** | **0.759** | **0.722** |
| Cornell | 0.742 | 0.719 | 0.676 | 0.639 | **0.759** | **0.736** | **0.693** | **0.656** |
| Wisconsin | 0.859 | 0.841 | 0.804 | 0.767 | **0.873** | **0.855** | **0.818** | **0.781** |

## 4.2 Assessing the GRDP in molecular graphs

The rationale for choosing this specific assortment of datasets is to demonstrate the capability of the decoder within an environment characterized by multiple small graphs, a scenario frequently encountered in the field of chemistry where such tasks are gaining prominence.

The rising importance of these tasks is evident in advancements such as the development of proteolysis-targeting chimera (PROTAC) molecules (Békés et al., 2022). Such a task can readily be reformulated as a link prediction problem, as a critical phase in the development of PROTACs involves the integration of two molecules via a linker, which is another molecular fragment. Given the scarce availability of PROTAC molecules, it was deemed fit to simulate this task using link prediction within the molecules of both our datasets.

Moreover, substructure identification is a pervasive problem across drug development, where small changes in their molecular motifs can have huge impacts across molecular properties (Klekota & Roth, 2008). Being able to address this problem from an unsupervised approach offers advantages regarding generalization towards unseen substructures. With this purpose in mind, we convert this problem into a node clustering task, where the clustering labels utilized were the node atomic numbers, aiming to ascertain whether a superior apprehension of the graph topology facilitates the clustering of chemically analogous nodes.

As observed in Table 3, our model distinctly outperforms the established baselines. This underscores the model's enhanced efficacy in deciphering the inherent topology of the graphs and its competency to rationalize over unseen small graphs.

Our findings indicate improvements in both architectures, thereby reinforcing the supposition that the GRDPG decoder enhances the comprehension of more intricate graph topologies (see Table 4). In this scenario, both the applicability and versatility of the model are pivotal, providing insights that are not only theoretically significant but also relevant in practical real-world chemical contexts.

Table 3: Link Prediction in Molecular Graphs

| Dataset | GAE | | GAE + GRDP | | VGAE | | VGAE + GRDP | |
|---|---|---|---|---|---|---|---|---|
| | AUC | AP | AUC | AP | AUC | AP | AUC | AP |
| QM9 | 0.914 | 0.882 | **0.959** | **0.936** | **0.959** | **0.937** | 0.903 | 0.869 |
| ZINC | 0.848 | 0.805 | **0.879** | **0.841** | 0.842 | 0.798 | **0.868** | **0.827** |

Table 4: Node Clustering in Molecular Graphs

(a) GAE Architecture Results

| Dataset | GAE | | | | GAE + GRDP | | | |
|---|---|---|---|---|---|---|---|---|
| | acc | NMI | F1 | adj-RI | acc | NMI | F1 | adj-RI |
| QM9 | **0.148** | 0.164 | 0.200 | 0.066 | 0.146 | **0.534** | **0.201** | **0.353** |
| ZINC | **0.183** | 0.243 | **0.235** | 0.148 | 0.177 | **0.267** | 0.228 | **0.170** |

(b) VGAE Architecture Results

| Dataset | VGAE | | | | VGAE + GRDP | | | |
|---|---|---|---|---|---|---|---|---|
| | acc | NMI | F1 | adj-RI | acc | NMI | F1 | adj-RI |
| QM9 | 0.153 | 0.167 | 0.206 | 0.072 | **0.156** | **0.423** | **0.207** | **0.269** |
| ZINC | 0.187 | 0.243 | 0.240 | 0.135 | **0.192** | **0.267** | **0.249** | **0.166** |

## 5 Conclusions

Overall, in this paper we identify a seemingly unexplored direction of research into graph decoding, and propose motivations and empirical results as of why it needs further work from the community.

We propose a novel graph decoding model that alleviates some of the assumptions made by previous approaches, which we obtain good empirical performance with. Furthermore we also define a novel benchmark that aligns link prediction and node clustering problems to different real world scenarios.

This work can be extended along several possible directions. Firstly, defining the $q$ hyperparameter from a less empirical point of view might help us better understand the relations needed for capturing different graph topologies, so further theoretical work is needed within this framework. Secondly, similar ideas could be adopted by the knowledge graph community, where more complex and less constrained scoring functions could bring benefits towards more powerful entity and relationship embeddings. Lastly, there have been other generative models that have been proposed for alleviating the previously outlined limitation, such is the case of graph root distribution model of Lei (2020). Further benchmarking this novel family of decoders could help bring further insight into the graph decoding and generation process.

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

# A  EXAMPLES OF NEGATIVE EIGENVALUE EXISTENCE

## A.1  NON-BIPARTITE CYCLIC GRAPH

Consider a non-bipartite graph with vertices $A$, $B$, and $C$, forming a triangle. The adjacency matrix, $A$, for this graph is given by:

$$A = \begin{bmatrix} 0 & 1 & 1 \\ 1 & 0 & 1 \\ 1 & 1 & 0 \end{bmatrix}.$$

To find a specific eigenvector, $v = \begin{bmatrix} 1 \\ x \\ y \end{bmatrix}$, for a corresponding eigenvalue, $\lambda$, we substitute into $Av = \lambda v$, yielding the system:

$$x + y = \lambda, \quad 1 + y = x\lambda, \quad 1 + x = y\lambda.$$

Solving for $x$, we get:

$$x = \frac{\lambda - 1}{\lambda^2 - 1}.$$

The characteristic polynomial, $P(\lambda) = \det(A - \lambda I)$, results in

$$-\lambda(\lambda - \sqrt{3})(\lambda + \sqrt{3}) = 0.$$

Hence, the eigenvalues are $\lambda = 0$, $\lambda = \sqrt{3}$, and $\lambda = -\sqrt{3}$.

Substituting the negative eigenvalue, $\lambda = -\sqrt{3}$, back into the equation for $x$, we find:

$$x = -\frac{\sqrt{3} + 1}{2}.$$

And then, substituting back into $y = x\lambda - 1$, yields:

$$y = \frac{4 + \sqrt{3}}{2}.$$

Thus, the specific eigenvector corresponding to the negative eigenvalue is:

$$v = \begin{bmatrix} 1 \\ -\frac{\sqrt{3}+1}{2} \\ \frac{4+\sqrt{3}}{2} \end{bmatrix}.$$

This example illustrates the existence of a negative eigenvalue, $\lambda = -\sqrt{3}$, in the adjacency matrix of a non-bipartite graph, and the construction of the corresponding eigenvector, $[1, -\frac{\sqrt{3}+1}{2}, \frac{4+\sqrt{3}}{2}]$.

## A.2  6-VERTEX NON-BIPARTITE GRAPH

Consider a non-bipartite graph with six vertices, represented by the adjacency matrix,

$$A = \begin{bmatrix} 0 & 1 & 0 & 0 & 0 & 1 \\ 1 & 0 & 1 & 0 & 0 & 0 \\ 0 & 1 & 0 & 1 & 0 & 1 \\ 0 & 0 & 1 & 0 & 1 & 0 \\ 0 & 0 & 0 & 1 & 0 & 1 \\ 1 & 0 & 1 & 0 & 1 & 0 \end{bmatrix}.$$

Employing numerical computation, we uncover three negative eigenvalues for the matrix, namely $\lambda_1 \approx -2.414$, $\lambda_2 \approx -1$, and $\lambda_3 \approx -0.414$, each associated with corresponding eigenvectors:

$$v_1 \approx \begin{bmatrix} -0.354 \\ 0.354 \\ -0.5 \\ 0.354 \\ -0.354 \\ 0.5 \end{bmatrix}, \quad v_2 \approx \begin{bmatrix} 0.5 \\ -0.5 \\ 0 \\ 0.5 \\ -0.5 \\ 0 \end{bmatrix}, \quad v_3 \approx \begin{bmatrix} -0.354 \\ -0.354 \\ 0.5 \\ -0.354 \\ -0.354 \\ 0.5 \end{bmatrix}.$$

The discerned eigenvalues and their associated eigenvectors confirm the presence of negative eigenvalues in the adjacency matrix of complex non-bipartite graphs, with the eigenvectors shedding light on the nodes' connection within the non-bipartite substructures.

## B  EXPERIMENTS FULL RESULTS

In the following section we have a full compilation of all the results of our experiments. The following tables include the results for the inner product decoder benchmarked against a wide range of possible values of $q$ in the GRDPG decoder. In them, the hyperparameter $q$ represents the amount of negative ones in proportion to the positive ones , $p$, that populate the diagonal of the $I_{p,q}$ matrix. Thus if the embedding space has dimension 2 and q is $\frac{1}{2}$, the matrix $I_{p,q}$ would be comprised of one of each, 1 and -1, on its diagonal.

Therefore, the $q$ values listed as 0 are referring to the original inner product decoder, or RDPG, as the $I_{p,q}$ matrix becomes the identity matrix $I$ and the GRDPG graph becomes the dot product between latent space representations of the nodes, the inner product decoder.

### B.1  LINK PREDICTION

Table 5: Results for Link Prediction of Cora

| | GAE | | VGAE | |
|---|---|---|---|---|
| $q$ | AUC | AP | AUC | AP |
| 0 | $0.937 \pm 0.005$ | $0.944 \pm 0.003$ | $0.935 \pm 0.006$ | $0.936 \pm 0.005$ |
| $\frac{1}{16}$ | $0.906 \pm 0.006$ | $0.911 \pm 0.011$ | $0.897 \pm 0.009$ | $0.899 \pm 0.011$ |
| $\frac{1}{8}$ | $0.898 \pm 0.012$ | $0.904 \pm 0.009$ | $0.890 \pm 0.010$ | $0.895 \pm 0.009$ |
| $\frac{1}{4}$ | $0.895 \pm 0.007$ | $0.901 \pm 0.006$ | $0.882 \pm 0.009$ | $0.889 \pm 0.007$ |
| $\frac{1}{2}$ | $0.866 \pm 0.011$ | $0.876 \pm 0.011$ | $0.856 \pm 0.011$ | $0.865 \pm 0.016$ |

Table 6: Results for Link Prediction of Citeseer

| | GAE | | VGAE | |
|---|---|---|---|---|
| $q$ | AUC | AP | AUC | AP |
| 0 | $0.921 \pm 0.008$ | $0.933 \pm 0.007$ | $0.930 \pm 0.007$ | $0.937 \pm 0.006$ |
| $\frac{1}{16}$ | $0.900 \pm 0.006$ | $0.911 \pm 0.005$ | $0.888 \pm 0.022$ | $0.899 \pm 0.018$ |
| $\frac{1}{8}$ | $0.894 \pm 0.005$ | $0.904 \pm 0.004$ | $0.881 \pm 0.017$ | $0.892 \pm 0.015$ |
| $\frac{1}{4}$ | $0.880 \pm 0.009$ | $0.895 \pm 0.010$ | $0.866 \pm 0.015$ | $0.882 \pm 0.012$ |
| $\frac{1}{2}$ | $0.839 \pm 0.024$ | $0.866 \pm 0.017$ | $0.799 \pm 0.034$ | $0.831 \pm 0.027$ |

Table 7: Results for Link Prediction of Wisconsin

| | GAE | | VGAE | |
|---|---|---|---|---|
| $q$ | AUC | AP | AUC | AP |
| 0 | $0.808 \pm 0.028$ | $0.841 \pm 0.027$ | $0.836 \pm 0.041$ | $0.861 \pm 0.043$ |
| $\frac{1}{16}$ | $0.793 \pm 0.038$ | $0.826 \pm 0.049$ | $0.763 \pm 0.076$ | $0.804 \pm 0.064$ |
| $\frac{1}{8}$ | $0.815 \pm 0.064$ | $0.843 \pm 0.070$ | $0.812 \pm 0.043$ | $0.849 \pm 0.040$ |
| $\frac{1}{4}$ | $0.786 \pm 0.028$ | $0.827 \pm 0.018$ | $0.753 \pm 0.032$ | $0.794 \pm 0.052$ |
| $\frac{1}{2}$ | $0.700 \pm 0.099$ | $0.766 \pm 0.074$ | $0.693 \pm 0.074$ | $0.726 \pm 0.074$ |

Table 8: Results for Link Prediction of Cornell

| | GAE | | VGAE | |
|---|---|---|---|---|
| $q$ | AUC | AP | AUC | AP |
| 0 | $0.701 \pm 0.128$ | $0.779 \pm 0.085$ | $0.768 \pm 0.082$ | $0.817 \pm 0.063$ |
| $\frac{1}{16}$ | $0.791 \pm 0.035$ | $0.837 \pm 0.014$ | $0.751 \pm 0.032$ | $0.788 \pm 0.036$ |
| $\frac{1}{8}$ | $0.796 \pm 0.070$ | $0.844 \pm 0.060$ | $0.758 \pm 0.027$ | $0.794 \pm 0.033$ |
| $\frac{1}{4}$ | $0.726 \pm 0.071$ | $0.778 \pm 0.087$ | $0.780 \pm 0.113$ | $0.791 \pm 0.113$ |
| $\frac{1}{2}$ | $0.707 \pm 0.049$ | $0.788 \pm 0.038$ | $0.751 \pm 0.114$ | $0.802 \pm 0.092$ |

Table 9: Results for Link Prediction of Texas

| | GAE | | VGAE | |
|---|---|---|---|---|
| $q$ | AUC | AP | AUC | AP |
| 0 | $0.682 \pm 0.138$ | $0.750 \pm 0.115$ | $0.805 \pm 0.018$ | $0.856 \pm 0.019$ |
| $\frac{1}{16}$ | $0.633 \pm 0.129$ | $0.725 \pm 0.097$ | $0.759 \pm 0.046$ | $0.812 \pm 0.050$ |
| $\frac{1}{8}$ | $0.706 \pm 0.119$ | $0.775 \pm 0.099$ | $0.705 \pm 0.123$ | $0.779 \pm 0.098$ |
| $\frac{1}{4}$ | $0.707 \pm 0.057$ | $0.791 \pm 0.056$ | $0.728 \pm 0.058$ | $0.782 \pm 0.056$ |
| $\frac{1}{2}$ | $0.649 \pm 0.122$ | $0.745 \pm 0.108$ | $0.669 \pm 0.091$ | $0.753 \pm 0.083$ |

Table 10: Results for Link Prediction of QM9

| | GAE | | VGAE | |
|---|---|---|---|---|
| $q$ | AUC | AP | AUC | AP |
| 0 | $0.914 \pm 0.001$ | $0.882 \pm 0.001$ | $0.903 \pm 0.010$ | $0.869 \pm 0.011$ |
| $\frac{1}{16}$ | $0.950 \pm 0.003$ | $0.918 \pm 0.004$ | $0.950 \pm 0.003$ | $0.919 \pm 0.004$ |
| $\frac{1}{8}$ | $0.959 \pm 0.003$ | $0.936 \pm 0.004$ | $0.958 \pm 0.007$ | $0.934 \pm 0.009$ |
| $\frac{1}{4}$ | $0.957 \pm 0.002$ | $0.934 \pm 0.004$ | $0.959 \pm 0.004$ | $0.937 \pm 0.006$ |
| $\frac{1}{2}$ | $0.956 \pm 0.003$ | $0.933 \pm 0.005$ | $0.957 \pm 0.005$ | $0.935 \pm 0.008$ |

Table 11: Results for Link Prediction of ZINC

| | GAE | | VGAE | |
|---|---|---|---|---|
| $q$ | AUC | AP | AUC | AP |
| 0 | $0.848 \pm 0.008$ | $0.805 \pm 0.009$ | $0.842 \pm 0.017$ | $0.798 \pm 0.020$ |
| $\frac{1}{16}$ | $0.875 \pm 0.004$ | $0.831 \pm 0.005$ | $0.866 \pm 0.009$ | $0.822 \pm 0.010$ |
| $\frac{1}{8}$ | $0.879 \pm 0.005$ | $0.841 \pm 0.008$ | $0.868 \pm 0.010$ | $0.827 \pm 0.011$ |
| $\frac{1}{4}$ | $0.873 \pm 0.006$ | $0.833 \pm 0.008$ | $0.868 \pm 0.000$ | $0.826 \pm 0.001$ |
| $\frac{1}{2}$ | $0.876 \pm 0.006$ | $0.837 \pm 0.009$ | $0.868 \pm 0.000$ | $0.827 \pm 0.001$ |

## B.2 NODE CLUSTERING.

Table 12: Results for Node Clustering of Cora

| | GAE | | | | |
|---|---|---|---|---|---|
| $q$ | acc | NMI | F1 | precision | adj-RI |
| 0 | $0.212 \pm 0.112$ | $0.485 \pm 0.015$ | $0.218 \pm 0.129$ | $0.249 \pm 0.154$ | $0.421 \pm 0.028$ |
| $\frac{1}{16}$ | $0.092 \pm 0.081$ | $0.494 \pm 0.019$ | $0.099 \pm 0.091$ | $0.116 \pm 0.109$ | $0.427 \pm 0.034$ |
| $\frac{1}{8}$ | $0.316 \pm 0.107$ | $0.472 \pm 0.027$ | $0.314 \pm 0.111$ | $0.343 \pm 0.126$ | $0.404 \pm 0.037$ |
| $\frac{1}{4}$ | $0.218 \pm 0.073$ | $0.460 \pm 0.026$ | $0.233 \pm 0.071$ | $0.264 \pm 0.079$ | $0.387 \pm 0.040$ |
| $\frac{1}{2}$ | $0.171 \pm 0.029$ | $0.387 \pm 0.034$ | $0.182 \pm 0.046$ | $0.226 \pm 0.080$ | $0.294 \pm 0.032$ |
| | VGAE | | | | |
| $q$ | acc | NMI | F1 | precision | adj-RI |
| 0 | $0.190 \pm 0.104$ | $0.504 \pm 0.021$ | $0.198 \pm 0.113$ | $0.221 \pm 0.131$ | $0.453 \pm 0.013$ |
| $\frac{1}{16}$ | $0.205 \pm 0.089$ | $0.471 \pm 0.031$ | $0.213 \pm 0.077$ | $0.254 \pm 0.067$ | $0.383 \pm 0.042$ |
| $\frac{1}{8}$ | $0.110 \pm 0.093$ | $0.462 \pm 0.036$ | $0.112 \pm 0.101$ | $0.136 \pm 0.126$ | $0.363 \pm 0.049$ |
| $\frac{1}{4}$ | $0.222 \pm 0.077$ | $0.443 \pm 0.021$ | $0.218 \pm 0.076$ | $0.256 \pm 0.107$ | $0.344 \pm 0.042$ |
| $\frac{1}{2}$ | $0.077 \pm 0.017$ | $0.391 \pm 0.048$ | $0.068 \pm 0.018$ | $0.074 \pm 0.026$ | $0.276 \pm 0.049$ |

Table 13: Results for Node Clustering of CiteSeer

| | | GAE | | | |
|---|---|---|---|---|---|
| $q$ | acc | NMI | F1 | precision | adj-RI |
| 0 | $0.165 \pm 0.052$ | $0.268 \pm 0.034$ | $0.172 \pm 0.057$ | $0.202 \pm 0.066$ | $0.215 \pm 0.047$ |
| $\frac{1}{16}$ | $0.159 \pm 0.088$ | $0.333 \pm 0.024$ | $0.163 \pm 0.098$ | $0.178 \pm 0.115$ | $0.310 \pm 0.039$ |
| $\frac{1}{8}$ | $0.182 \pm 0.110$ | $0.332 \pm 0.013$ | $0.192 \pm 0.119$ | $0.211 \pm 0.135$ | $0.297 \pm 0.017$ |
| $\frac{1}{4}$ | $0.192 \pm 0.082$ | $0.290 \pm 0.033$ | $0.191 \pm 0.085$ | $0.199 \pm 0.091$ | $0.244 \pm 0.041$ |
| $\frac{1}{2}$ | $0.169 \pm 0.046$ | $0.268 \pm 0.044$ | $0.171 \pm 0.055$ | $0.202 \pm 0.070$ | $0.217 \pm 0.048$ |
| | | VGAE | | | |
| $q$ | acc | NMI | F1 | precision | adj-RI |
| 0 | $0.221 \pm 0.106$ | $0.325 \pm 0.031$ | $0.221 \pm 0.106$ | $0.241 \pm 0.114$ | $0.308 \pm 0.043$ |
| $\frac{1}{16}$ | $0.148 \pm 0.043$ | $0.310 \pm 0.018$ | $0.156 \pm 0.045$ | $0.176 \pm 0.060$ | $0.275 \pm 0.029$ |
| $\frac{1}{8}$ | $0.189 \pm 0.104$ | $0.307 \pm 0.044$ | $0.194 \pm 0.111$ | $0.206 \pm 0.120$ | $0.272 \pm 0.061$ |
| $\frac{1}{4}$ | $0.130 \pm 0.042$ | $0.276 \pm 0.031$ | $0.129 \pm 0.046$ | $0.133 \pm 0.048$ | $0.226 \pm 0.034$ |
| $\frac{1}{2}$ | $0.172 \pm 0.067$ | $0.222 \pm 0.029$ | $0.173 \pm 0.075$ | $0.190 \pm 0.101$ | $0.170 \pm 0.031$ |

Table 14: Results for Node Clustering of Wisconsin

| | | GAE | | | |
|---|---|---|---|---|---|
| $q$ | acc | NMI | F1 | precision | adj-RI |
| 0 | $0.265 \pm 0.087$ | $0.129 \pm 0.043$ | $0.273 \pm 0.095$ | $0.356 \pm 0.097$ | $0.097 \pm 0.045$ |
| $\frac{1}{16}$ | $0.136 \pm 0.076$ | $0.137 \pm 0.057$ | $0.132 \pm 0.078$ | $0.230 \pm 0.053$ | $0.129 \pm 0.071$ |
| $\frac{1}{8}$ | $0.186 \pm 0.088$ | $0.134 \pm 0.050$ | $0.179 \pm 0.093$ | $0.243 \pm 0.108$ | $0.115 \pm 0.056$ |
| $\frac{1}{4}$ | $0.218 \pm 0.076$ | $0.123 \pm 0.048$ | $0.211 \pm 0.081$ | $0.279 \pm 0.057$ | $0.107 \pm 0.047$ |
| $\frac{1}{2}$ | $0.178 \pm 0.075$ | $0.107 \pm 0.031$ | $0.196 \pm 0.087$ | $0.300 \pm 0.107$ | $0.074 \pm 0.034$ |
| | | VGAE | | | |
| $q$ | acc | NMI | F1 | precision | adj-RI |
| 0 | $0.211 \pm 0.086$ | $0.153 \pm 0.041$ | $0.186 \pm 0.075$ | $0.261 \pm 0.091$ | $0.141 \pm 0.034$ |
| $\frac{1}{16}$ | $0.276 \pm 0.083$ | $0.137 \pm 0.048$ | $0.267 \pm 0.082$ | $0.325 \pm 0.078$ | $0.139 \pm 0.054$ |
| $\frac{1}{8}$ | $0.214 \pm 0.080$ | $0.112 \pm 0.027$ | $0.201 \pm 0.066$ | $0.260 \pm 0.033$ | $0.118 \pm 0.040$ |
| $\frac{1}{4}$ | $0.121 \pm 0.084$ | $0.121 \pm 0.037$ | $0.113 \pm 0.101$ | $0.181 \pm 0.128$ | $0.108 \pm 0.035$ |
| $\frac{1}{2}$ | $0.232 \pm 0.077$ | $0.082 \pm 0.014$ | $0.246 \pm 0.086$ | $0.311 \pm 0.099$ | $0.087 \pm 0.030$ |

Table 15: Results for Node Clustering of Texas

| | | GAE | | | |
|---|---|---|---|---|---|
| $q$ | acc | NMI | F1 | precision | adj-RI |
| 0 | $0.213 \pm 0.099$ | $0.082 \pm 0.014$ | $0.211 \pm 0.101$ | $0.316 \pm 0.073$ | $0.098 \pm 0.034$ |
| $\frac{1}{16}$ | $0.201 \pm 0.118$ | $0.098 \pm 0.018$ | $0.218 \pm 0.117$ | $0.357 \pm 0.104$ | $0.125 \pm 0.040$ |
| $\frac{1}{8}$ | $0.137 \pm 0.034$ | $0.087 \pm 0.019$ | $0.138 \pm 0.059$ | $0.271 \pm 0.060$ | $0.145 \pm 0.048$ |
| $\frac{1}{4}$ | $0.178 \pm 0.133$ | $0.088 \pm 0.026$ | $0.178 \pm 0.133$ | $0.245 \pm 0.132$ | $0.120 \pm 0.031$ |
| $\frac{1}{2}$ | $0.193 \pm 0.146$ | $0.078 \pm 0.022$ | $0.169 \pm 0.141$ | $0.219 \pm 0.136$ | $0.089 \pm 0.048$ |
| | | VGAE | | | |
| $q$ | acc | NMI | F1 | precision | adj-RI |
| 0 | $0.127 \pm 0.038$ | $0.122 \pm 0.026$ | $0.114 \pm 0.030$ | $0.206 \pm 0.087$ | $0.133 \pm 0.052$ |
| $\frac{1}{16}$ | $0.172 \pm 0.036$ | $0.122 \pm 0.035$ | $0.166 \pm 0.056$ | $0.267 \pm 0.051$ | $0.156 \pm 0.025$ |
| $\frac{1}{8}$ | $0.181 \pm 0.073$ | $0.103 \pm 0.017$ | $0.192 \pm 0.090$ | $0.283 \pm 0.125$ | $0.117 \pm 0.023$ |
| $\frac{1}{4}$ | $0.165 \pm 0.065$ | $0.141 \pm 0.032$ | $0.152 \pm 0.072$ | $0.247 \pm 0.115$ | $0.155 \pm 0.071$ |
| $\frac{1}{2}$ | $0.134 \pm 0.046$ | $0.097 \pm 0.038$ | $0.101 \pm 0.039$ | $0.177 \pm 0.063$ | $0.113 \pm 0.031$ |

Table 16: Results for Node Clustering of Cornell

| | | GAE | | | |
|---|---|---|---|---|---|
| $q$ | acc | NMI | F1 | precision | adj-RI |
| 0 | $0.226 \pm 0.101$ | $0.083 \pm 0.019$ | $0.212 \pm 0.088$ | $0.239 \pm 0.078$ | $0.066 \pm 0.007$ |
| $\frac{1}{16}$ | $0.189 \pm 0.065$ | $0.065 \pm 0.038$ | $0.172 \pm 0.080$ | $0.202 \pm 0.107$ | $0.052 \pm 0.030$ |
| $\frac{1}{8}$ | $0.165 \pm 0.069$ | $0.060 \pm 0.027$ | $0.148 \pm 0.077$ | $0.201 \pm 0.108$ | $0.042 \pm 0.023$ |
| $\frac{1}{4}$ | $0.205 \pm 0.070$ | $0.045 \pm 0.015$ | $0.161 \pm 0.073$ | $0.186 \pm 0.085$ | $0.033 \pm 0.028$ |
| $\frac{1}{2}$ | $0.208 \pm 0.072$ | $0.049 \pm 0.015$ | $0.174 \pm 0.080$ | $0.212 \pm 0.101$ | $0.045 \pm 0.019$ |
| | | VGAE | | | |
| $q$ | acc | NMI | F1 | precision | adj-RI |
| 0 | $0.166 \pm 0.066$ | $0.057 \pm 0.013$ | $0.161 \pm 0.069$ | $0.247 \pm 0.055$ | $0.043 \pm 0.017$ |
| $\frac{1}{16}$ | $0.189 \pm 0.063$ | $0.054 \pm 0.008$ | $0.169 \pm 0.086$ | $0.239 \pm 0.118$ | $0.038 \pm 0.028$ |
| $\frac{1}{8}$ | $0.233 \pm 0.082$ | $0.070 \pm 0.020$ | $0.219 \pm 0.094$ | $0.333 \pm 0.105$ | $0.041 \pm 0.018$ |
| $\frac{1}{4}$ | $0.154 \pm 0.031$ | $0.062 \pm 0.011$ | $0.123 \pm 0.052$ | $0.170 \pm 0.068$ | $0.049 \pm 0.019$ |
| $\frac{1}{2}$ | $0.179 \pm 0.039$ | $0.052 \pm 0.010$ | $0.166 \pm 0.043$ | $0.279 \pm 0.100$ | $0.035 \pm 0.016$ |

Table 17: Results for Node Clustering of QM9

| | | | GAE | | |
|---|---|---|---|---|---|
| $q$ | acc | NMI | F1 | precision | adj-RI |
| 0 | $0.148 \pm 0.002$ | $0.164 \pm 0.005$ | $0.200 \pm 0.002$ | $0.375 \pm 0.006$ | $0.066 \pm 0.003$ |
| $\frac{1}{16}$ | $0.157 \pm 0.004$ | $0.245 \pm 0.049$ | $0.207 \pm 0.005$ | $0.365 \pm 0.008$ | $0.117 \pm 0.038$ |
| $\frac{1}{8}$ | $0.151 \pm 0.005$ | $0.302 \pm 0.046$ | $0.198 \pm 0.005$ | $0.359 \pm 0.008$ | $0.163 \pm 0.032$ |
| $\frac{1}{4}$ | $0.146 \pm 0.013$ | $0.534 \pm 0.037$ | $0.197 \pm 0.016$ | $0.368 \pm 0.020$ | $0.353 \pm 0.036$ |
| $\frac{1}{2}$ | $0.151 \pm 0.014$ | $0.480 \pm 0.041$ | $0.201 \pm 0.018$ | $0.370 \pm 0.032$ | $0.327 \pm 0.046$ |
| | | | VGAE | | |
| $q$ | acc | NMI | F1 | precision | adj-RI |
| 0 | $0.153 \pm 0.004$ | $0.167 \pm 0.015$ | $0.206 \pm 0.005$ | $0.384 \pm 0.010$ | $0.072 \pm 0.012$ |
| $\frac{1}{16}$ | $0.156 \pm 0.006$ | $0.239 \pm 0.043$ | $0.207 \pm 0.008$ | $0.370 \pm 0.010$ | $0.109 \pm 0.034$ |
| $\frac{1}{8}$ | $0.151 \pm 0.007$ | $0.303 \pm 0.051$ | $0.197 \pm 0.009$ | $0.361 \pm 0.009$ | $0.158 \pm 0.041$ |
| $\frac{1}{4}$ | $0.138 \pm 0.004$ | $0.423 \pm 0.015$ | $0.185 \pm 0.005$ | $0.345 \pm 0.010$ | $0.267 \pm 0.027$ |
| $\frac{1}{2}$ | $0.144 \pm 0.006$ | $0.421 \pm 0.016$ | $0.194 \pm 0.008$ | $0.357 \pm 0.020$ | $0.269 \pm 0.017$ |

Table 18: Results for Node Clustering of ZINC

| | | | GAE | | |
|---|---|---|---|---|---|
| $q$ | acc | NMI | F1 | precision | adj-RI |
| 0 | $0.183 \pm 0.009$ | $0.243 \pm 0.006$ | $0.235 \pm 0.011$ | $0.500 \pm 0.024$ | $0.148 \pm 0.009$ |
| $\frac{1}{16}$ | $0.179 \pm 0.007$ | $0.260 \pm 0.011$ | $0.231 \pm 0.010$ | $0.493 \pm 0.016$ | $0.160 \pm 0.013$ |
| $\frac{1}{8}$ | $0.177 \pm 0.013$ | $0.260 \pm 0.008$ | $0.228 \pm 0.016$ | $0.498 \pm 0.019$ | $0.170 \pm 0.014$ |
| $\frac{1}{4}$ | $0.177 \pm 0.011$ | $0.267 \pm 0.009$ | $0.228 \pm 0.013$ | $0.501 \pm 0.014$ | $0.163 \pm 0.018$ |
| $\frac{1}{2}$ | $0.173 \pm 0.006$ | $0.261 \pm 0.009$ | $0.226 \pm 0.008$ | $0.498 \pm 0.017$ | $0.160 \pm 0.025$ |
| | | | VGAE | | |
| $q$ | acc | NMI | F1 | precision | adj-RI |
| 0 | $0.187 \pm 0.005$ | $0.243 \pm 0.006$ | $0.240 \pm 0.007$ | $0.514 \pm 0.026$ | $0.135 \pm 0.025$ |
| $\frac{1}{16}$ | $0.180 \pm 0.014$ | $0.264 \pm 0.007$ | $0.233 \pm 0.018$ | $0.503 \pm 0.017$ | $0.157 \pm 0.013$ |
| $\frac{1}{8}$ | $0.190 \pm 0.015$ | $0.264 \pm 0.005$ | $0.246 \pm 0.018$ | $0.516 \pm 0.024$ | $0.166 \pm 0.004$ |
| $\frac{1}{4}$ | $0.188 \pm 0.015$ | $0.267 \pm 0.000$ | $0.243 \pm 0.018$ | $0.513 \pm 0.024$ | $0.164 \pm 0.001$ |
| $\frac{1}{2}$ | $0.192 \pm 0.016$ | $0.267 \pm 0.001$ | $0.249 \pm 0.020$ | $0.528 \pm 0.019$ | $0.166 \pm 0.002$ |

## C  HYPERPARAMETERS REPORT

Table 19: Hyperparameters for Molecular Graph Datasets

| Hyperparameters | Values |
|---|---|
| Optimizer | Adam |
| Epochs | 200 |
| Activation Function | ReLU |
| Learning Rate | 0.0005 |
| Batch Size | 64 |
| Embedding Size | 16 |
| Splits QM9 | 80/10/10 |
| Splits Zinc | 88/10/2 |

Table 20: Hyperparameters for Non Molecular Datasets

| Hyperparameters | Values |
|---|---|
| Optimizer | Adam |
| Epochs | 200 |
| Activation Function | ReLU |
| Learning Rate | 0.001 |
| Embedding Size | 64 |
| Splits | 70/20/10 |

