# OpenReview forum: "Graph Decoding via Generalized Random Dot Product Graph"
_ICLR.cc/2024/Conference — Submitted to ICLR 2024_

### Official Review · Reviewer_hJLN · 2023-10-24

**Soundness:** 1 poor
**Presentation:** 1 poor
**Contribution:** 2 fair
**Rating:** 3
**Confidence:** 3

**Summary:**

This paper identifies the problem of the current GNN's inability to reconstruct a graph whose adjacency matrix has negative eigenvalues. It then proposes a method to manually negate several entries of the embedding vectors of nodes to allow negative eigenvalue. Experimental results show consistent improvement across datasets.

**Strengths:**

The identified problem is interesting and allowing models to generate an adjacency matrix with negative eigenvalues is a critical task. The authors explain the intuition behind the negative eigenvalue adjacency matrix well.

**Weaknesses:**

The presentation needs significant improvement. The use of bold/unbold letters to represent vectors is inconsistent, and some subscripts are missing.  While the core idea is interesting, it is poorly presented. As the authors emphasize the importance of the negative eigenvalues
and mention irregularity in graphs multiple times, there is no support for how often and when negative eigenvalues appear. (Though toy examples are provided in the appendix.) Some justification should be presented, either empirically obtained from data or theoretically quantified.

The proposed method is also without in-depth analysis. While the proposed matrix multiplication does allow negative eigenvalues, the effectiveness is unclear and never justified in the paper. Also, the negate matrix is manually picked, which is not flexible for more graphs (as the authors acknowledged in the conclusion).

**Questions:**

- The experiments only compared the proposed method with the backbone GNN. However, the advances recent GNN significantly outperforms GAE on various tasks. How does the proposed method compare to those methods.

- Does the generated matrix indeed have negative eigenvalues? If so, does the original matrix that the GNN is trained on also contain negative eigenvalues?

---

> ### Author Response · Authors · 2023-11-23
>
> Dear Reviewer, we thank you for your contribution to our work. We wanted to address some of the comments you left in your review.
>
> We purposely choose the vanilla implementations of such architectures since we believe that the proposed methodology only makes sense when compared to the traditional inner product decoder.
>
> Yes, both of them have negative eigenvalues. The adjacency matrix normalisation operations in GCNs do not alter the spectra of negative eigenvalues beyond their scale, and the generative model indeed generates negative eigenvalues, as stated in the text.

---

### Official Review · Reviewer_io5N · 2023-10-27

**Soundness:** 2 fair
**Presentation:** 3 good
**Contribution:** 1 poor
**Rating:** 3
**Confidence:** 4

**Summary:**

This work introduce a graph decoder by employing the Generalized Random Dot Product Graph (GRDPG) as a generative model for graph decoding. This methodology significantly enhances the performance of encoder-decoder architectures across a range of tasks.

**Strengths:**

1. Clear and straightforward illustraction of method.

**Weaknesses:**

1. Novelty is limited.  Equation (3), the center of the method, has been proposed in [1].

2. The meaning of negative eigenvalue in adjacency matrix remains unclear. Does it corresponds to some specific substructure or some graph statistics?

3. Uncertainties are not provided in experiment results.

4. Though this work proposes a decoder, generation tasks are not included in experiments.

[1] Patrick Rubin-Delanchy, Joshua Cape, Minh Tang, and Carey E. Priebe. A statistical interpretation of spectral embedding: the generalised random dot product graph. arXiv:1709.05506, 2021.

**Questions:**

1. Lorentz model in hyperbolic GNN [1] also uses a diagonal matrix with $\pm 1$. Could you please compare your Equation 3 with hyperbolic GNNs?

2. What is the detailed meaning of negative eigenvalues in graph? Some specific graph structures?

3. Can use approximate graph laplacian matrix ($L=D-A$) instead of adjacency matrix to avoid the positive-semi definite problem?

[1] Menglin Yang, Min Zhou, Zhihao Li, Jiahong Liu, Lujia Pan, Hui Xiong, Irwin King. Hyperbolic Graph Neural Networks: A Review of Methods and Applications. CoRR abs/2202.13852.

---

> ### Author Response · Authors · 2023-11-23
>
> Dear Reviewer, we thank you for your contribution to our work. We wanted to address some of the comments you left in your review.
>
> We were well aware of the contribution made by [1], we believe that even though our work is not catered around the development of the kernel, its application to a graph decoding process in the context of GAEs provides novelty to the field.
>
> We did not delve into the topological effects of the presence of negative eigenvalues, but it is clearly something we want to pursue shortly. Nevertheless we could informally link it with the presence of cycles and triangles.
>
> We believe that uncertainties are provided in the results tables in the form of measured deviations from the mean in the Supplementary Material. We will include them in the main text in future versions of the work.
>
> Thank you for your question regarding the comparison between our Equation 3 and the Lorentz model in hyperbolic GNNs. Our Equation 3, given by, provides a framework for computing the probability of an adjacency matrix
> Unlike the Lorentz model, which is specifically designed for hyperbolic space with its constant negative curvature, our model is more general. The diagonal matrix in our equation does not inherently assume a hyperbolic structure and could potentially be adapted to represent other types of geometric metrics, including Euclidean space.
> While the Lorentz model is adept at handling data with hierarchical structures characteristic of hyperbolic space, our model aims to be more versatile, allowing for adjustment of the diagonal matrix to cater to a wider range of graph topologies and data geometries. This adaptability makes our Equation 3 potentially applicable to a broader spectrum of graph-based learning tasks. It is important to note, however, that if it is specified to reflect hyperbolic geometry, our model could operate in a manner similar to the Lorentz model, thus serving as a specific case within our more general framework.
>
> About using a Laplacian. Indeed this is a valid approach, however, adopting this paradigm would heavily complicate the generative paradigm of the model. Implementing this model, would duplicate the target matrices needed, as the degree matrix D would also need to be generated for the adjacency matrix reconstruction. We consider this to be out of the scope of our proposed article, however, we will consider this further for future work.

---

### Official Review · Reviewer_ggnG · 2023-10-31

**Soundness:** 2 fair
**Presentation:** 1 poor
**Contribution:** 1 poor
**Rating:** 1
**Confidence:** 4

**Summary:**

This paper introduces a modification to the decoder, i.e. the dot product of the existing graph encoder-decoder framework called GRDPG. It intends to empower the decoder to represent the adjacency with negative eigenvalues. Experiments show some improvement.

**Strengths:**

- The proposed method can be flexibly applied on any existing dot-product-based reconstruction functions.
- Experiments show some improvement somehow.

**Weaknesses:**

- The paper is skeptically to be considered as lacking good structure, especially the connection between the dot product and its deficiency of presenting negative eigenvalues.
- Very often the sentences are repeated, presenting the same meaning, e.g. the first two paragraphs on page 2, the fourth paragraph on page 3, and so on.
- Typos are quite frequent, e.g. the third equation $p(Z|X,A \$ (also there is no index for all equations or formulas except those in the method), $Z=z_i^Tz_j$, and so on.
- The experimental results are not convincing, especially on the structure perception link prediction task, and also the random seeds and variances are missing to report.
- Overall, the quality of the paper is far below the requirements of ICLR.

**Questions:**

- My first question is that it would be nice to make sure that the existing papers that hold the semi-positive defined assumption of the probability of reconstruction is entailed by its assumption of undirected graphs.
- It is highly recommended to further investigate which type of $I_{p,q}$ works best for different graph structures.
- Why is node clustering an appropriate experiment to verify the model? Should it be more related to the task of graph structure perception?

---

> ### Author Response · Authors · 2023-11-23
>
> Dear Reviewer, we thank you for your contribution to our work. We wanted to address some of the comments you left in your review.
>
> In future submissions, we will address the relationship between graph structure and the type of I. We see that elucidation of this point can bring value to the presented work.
>
> Node clustering is a standard task for unsupervised graph representation learning. The objective of this task is to showcase the increased capability of algorithms in capturing the topology of the graph, trying to reproduce node labels without explicitly optimising for them. For further information the reviewer can check the following paper: Pan, S., Hu, R., Long, G., Jiang, J., Yao, L., & Zhang, C. (2018). Adversarially Regularized Graph Autoencoder for Graph Embedding. arXiv:1802.04407

---

### Official Review · Reviewer_AHQZ · 2023-11-02

**Soundness:** 2 fair
**Presentation:** 1 poor
**Contribution:** 2 fair
**Rating:** 1
**Confidence:** 5

**Summary:**

In this paper, the authors propose to use a GRDPG-based decoder, instead of the classical RDPG or inner-product decoder, in the framework of GAEs and GVAEs.

The introduction of the paper is very good. However, the rest of the paper is very repetitive, and the important details are missing.

I think that four pages to describe the impact of negative eigenvalues is too much. And the paper jumps from the description of the GRDPG (Section 3) to the experiments (Section 4), without explaining the proposed method, architecture, or something.

**Strengths:**

The introduction is very good, as said before. And the idea of using GRDPGs as decoder architecture is interesting, but I think it needs more work.

**Weaknesses:**

Throughout the paper, the authors insist on non-bipartite graphs, as if it was a requirement or something. Even for showing the existence of graphs with negative eigenvalues, while it would have made much more sense to say, "bipartite graphs have spectra of $A$ symmetric around 0".

In this sense, Section 2.2 (and Appendix A) are not needed at all.

The GRPDG method requires knowing in advance the number of positive/negative eigenvalues, so it's very important to know how the authors addressed this. Is it always the same? Is it chosen for each dataset? How?
If I guessed correctly, the authors try several different values of $q$ (for each dataset maybe?) and keep the result with the best performance.
The lack of information on this crucial part is not acceptable.

**Questions:**

In the preliminary section, the description for the GAE loss functions says the expectation with respect to some q, which is not defined before.

Minor comments:
 - the equations are part of the text. The punctuation after the equation is missing in general.

---

> ### Author Response · Authors · 2023-11-23
>
> Dear Reviewer, we thank you for your contribution to our work. We wanted to address some of the comments you left in your review.
>
> We stress the fact that bipartite graphs, by definition, are a case that fulfils the proposed requirements for negative eigenvalues even though they are not subject of study in this work purposely.
> The value of q, the proportion of negative eigenvalues, is searched by greedy search of the parameter per residue. This work does not establish any relationship between q and the data.
>
> The equation defining the GAE is extracted from Kipf, T. N., & Welling, M. (2016). Variational Graph Auto-Encoders. Advances in Neural Information Processing Systems. arXiv:1611.07308 (2016). There, in eq. (1) is stated q as q(Z | X, A) = QN i=1 q(zi | X, A), with q(zi | X, A) = N (zi | µi , diag(σ 2 i )).

---

### Meta-Review · Area_Chair_bbTJ · 2023-12-05

**Metareview:**

This paper presents an interesting approach for graph decoding. However, reviewers unanimously called into doubt issues around the work's novelty, presentation, and various other issues. These issues persisted after the Authors' rebuttal, and therefore I must recommend rejection at this time.

**Justification For Why Not Higher Score:**

There is a general consensus between all reviewers that major work is needed before this work would pass the bar for ICLR. The Authors' rebuttal did not change this stance.

**Justification For Why Not Lower Score:**

N/A

---

### Decision · Program_Chairs · 2024-01-16

Reject